# Inside the Insulin Secretory Granule

**DOI:** 10.3390/metabo11080515

**Published:** 2021-08-05

**Authors:** Mark Germanos, Andy Gao, Matthew Taper, Belinda Yau, Melkam A. Kebede

**Affiliations:** School of Medical Sciences, Faculty of Medicine and Health, Charles Perkins Centre, University of Sydney, Camperdown, Sydney 2006, Australia; mark.germanos@sydney.edu.au (M.G.); agao9724@uni.sydney.edu.au (A.G.); mtap8540@uni.sydney.edu.au (M.T.); belinda.yau@sydney.edu.au (B.Y.)

**Keywords:** insulin, islet amyloid polypeptide (IAPP), granin, secretory pathway, trans-Golgi network (TGN), granule, pancreatic β-cell

## Abstract

The pancreatic β-cell is purpose-built for the production and secretion of insulin, the only hormone that can remove glucose from the bloodstream. Insulin is kept inside miniature membrane-bound storage compartments known as secretory granules (SGs), and these specialized organelles can readily fuse with the plasma membrane upon cellular stimulation to release insulin. Insulin is synthesized in the endoplasmic reticulum (ER) as a biologically inactive precursor, proinsulin, along with several other proteins that will also become members of the insulin SG. Their coordinated synthesis enables synchronized transit through the ER and Golgi apparatus for congregation at the *trans*-Golgi network, the initiating site of SG biogenesis. Here, proinsulin and its constituents enter the SG where conditions are optimized for proinsulin processing into insulin and subsequent insulin storage. A healthy β-cell is continually generating SGs to supply insulin in vast excess to what is secreted. Conversely, in type 2 diabetes (T2D), the inability of failing β-cells to secrete may be due to the limited biosynthesis of new insulin. Factors that drive the formation and maturation of SGs and thus the production of insulin are therefore critical for systemic glucose control. Here, we detail the formative hours of the insulin SG from the luminal perspective. We do this by mapping the journey of individual members of the SG as they contribute to its genesis.

## 1. Introduction

The insulin secretory granule (SG) in the pancreatic β-cell is essential for glucose homeostasis in the body. It is both the site of proinsulin conversion into insulin and C-peptide [1], as well as the storage compartment for mature insulin to be readily available for secretion upon nutrient stimuli. Insulin is first synthesized as pre-proinsulin at the endoplasmic reticulum (ER), immediately converted to proinsulin, and transported through the Golgi to the trans-Golgi network (TGN). Here, proinsulin, along with other cargo proteins, is partitioned and sorted into its destination compartment, the immature SG (ISG) [1]. In the ISG, at least 99% of proinsulin is ultimately converted to insulin and C-peptide in a 1:1 molar ratio via proteolytic cleavages by the proprotein convertases PC1/3 and PC2 [2,3,4,5]. This coincides with several processes that facilitate SG maturation, including luminal acidification [6], selective removal of certain soluble components [7], and Zn^2+^-mediated insulin crystallization [8]. Finally, in response to nutrient stimuli, these mature SGs (MSGs) are mobilized to fuse with the plasma membrane and deliver insulin to the bloodstream.

Importantly, ISGs can also undergo regulated secretion [9], which can be heightened in situations of increased β-cell demand [10,11,12] and may explain the higher circulating proinsulin to insulin ratio observed in both pre-diabetic and diabetic patients [13,14,15,16,17,18,19]. The mechanism behind increased proinsulin secretion is unknown; although it has been suggested to result from defective proinsulin trafficking or processing, and/or the premature release of ISGs [20,21]. Interestingly, β-cells from animal models of type 2 diabetes (T2D) display a compensatory expansion of the secretory pathway, characterized by increased proinsulin biogenesis but exhibit a thorough depletion of MSGs, pointing to the existence of a bottleneck in the secretory pathway resulting in an MSG replenishment defect during β-cell failure [12]. Therefore, there is a diversion away from SG maturation in favor of ISG secretion, limiting the compensatory capacity of the β-cell during metabolic stress.

Alongside insulin, the β-cell SG contains a cocktail of cargo proteins. These proteins drive trafficking through the regulated secretory pathway and are also released to affect systemic function [22,23,24,25]. Luminal enzymes accompany the cargo from synthesis in the ER through to storage in the MSG but are under tight regulation to restrict their activity to the correct site [4]. The ionic composition of the lumen controls protein behavior and is generated by a range of transmembrane channels and transporters that are stationed throughout the secretory pathway [26,27,28,29,30,31]. Finally, sorting receptors can escort unwanted components away from the maturing SG to refine its contents after formation [7,32]. In this review, we will explore the major luminal components of the β-cell SG. These components will be discussed in relation to secretory pathway dysfunction, providing context to critical aspects of β-cell failure. However, first, we will start with a historical overview of the process of insulin SG formation.

### Historical Overview of Insulin SG Formation

Pioneering efforts in the 1980s elucidated the main concepts surrounding β-cell granule biogenesis. Orci first used immunogold labelling of total insulin, with an antibody that recognizes both proinsulin and insulin, to show that it is closely associated with membranes of the Golgi apparatus until the TGN, where it dissociates and concentrates into a mildly condensing core [33]. This core buds from the TGN into clathrin-coated ISGs, which develop into non-clathrin coated MSGs [33]. At the time, Halban was using pulse-chase methods to incorporate radiolabeled arginine and lysine analogues into newly synthesized proinsulin to inhibit its post-translational processing into insulin [9]. In collaboration, they inhibited proinsulin conversion and utilized autoradiography with clathrin-immunolabeling to provide the first direct evidence that proinsulin traffics from the TGN into clathrin-coated ISGs before its conversion into insulin and C-peptide [1]. Moreover, proinsulin conversion was shown to be required for complete SG maturation, as these analogue-treated cells could not form an electron-dense core which is characteristic of the MSG [1]. Indeed, the development of proinsulin and insulin-specific monoclonal antibodies later confirmed that proinsulin localization is most concentrated in the ISG compartment while insulin dominates the MSG compartment [6].

In 1987, Rhodes and Halban released a landmark study using radiolabeled proinsulin to follow the efficiency of its trafficking and conversion and the events of β-cell SG exocytosis [2]. The study found that 99% of proinsulin entered ISGs to lend itself for conversion, and that the resulting newly synthesized SGs were preferentially secreted over older SGs when exposed to glucose. Importantly, Halban had already shown that radiolabeled conversion-resistant proinsulin is released from the β-cell at the same rate as the non-resistant radiolabeled insulin product, therefore demonstrating that the ISGs housing proinsulin are also secretion-competent [9]. Collectively, foundational work from the 1980s suggested that the SG is the minimal functional unit for exocytosis, is formed through stringent processes, and is endowed with factors required for its regulated release early after formation. It would follow that delayed MSG production could result in the increased release of ISGs and hyperproinsulinemia, and thus a failure of the β-cell to respond to glucose with the secretion of insulin [12,34,35]. Ensuing efforts centering on answering how these carriers are formed have found that an ordered system of ionic and molecular factors underlie how SG proteins are sorted, packaged, and processed [36]. Likewise, efforts centered around understanding the preferential nature of exocytosis have facilitated the characterization of a vast network of components which confer mobility and fusion-competence to prepare the SG for release [37].

The MSG holds at least 50 unique soluble and transmembrane proteins [38], and the biosynthesis of many are thought to be commonly regulated at the translational level following exposure of the β-cell to glucose [39]. This enables their synchronized transit and congregation at the TGN, but from here, several proteins will traverse the ISG compartment on their way to other destinations. Due to this, the ISG intermediary was once the centerpiece of debate concerning the mode of transport that proinsulin and other regulated secretory proteins take en route to the MSG [40]. In the early 1990s, Arvan and colleagues found that C-peptide (the fragment generated from proinsulin after its complete conversion) could be released from the β-cell in molar excess to that of insulin during non-stimulatory conditions [41]. Follow up studies characterized the kinetics of this ‘*constitutive-like*’ secretion, specifically showing that this pathway emanates from the SG compartment and temporally coincides with the maturation of ISGs into MSGs [42]. Subsequent demonstration that insulin, but not proinsulin, is capable of forming insoluble hexamers, led to the idea that insulin condensation within the core of the SG permits the excursion of C-peptide out of the maturing granule as the soluble fraction is removed [43]. Moreover, by analyzing the regulated secretion of lysosomal hydrolase cathepsin B at different time points following pulse-chase radiolabeling, it was revealed that pro-cathepsin B entered the ISG only to be removed from the ISG shortly after entry [43]. Taken together, these studies established the presence of post-Golgi sorting mechanisms that serve to facilitate SG maturation by refining its composition.

Arvan thus proposed that members of the SG were not exclusively trafficked into the regulated secretory pathway from the TGN, but rather that an assortment of proteins were delivered into ISGs through means of unregulated ‘*bulk-flow*’—largely due to the stoichiometric infeasibility of sorting receptors existing for each cargo [44]. Subsequent post-Golgi mechanisms served to remove and traffic non-regulated secretory proteins to other destinations and drive the maturation of the SG. The term ‘*sorting by retention*’ was used to describe the selective condensation of proteins within the maturing SG, and ‘*sorting by exit*’ was used to describe budding from the vesicle that sequesters parts of the soluble fraction to remove other proteins [44] (Figure 1). This proposal sparked a debate; in particular, proponents against *bulk-flow* asserted that entry of proinsulin and other key granular components into the ISG could not be through a passive, unregulated mechanism [45]. In the end, the field came to the consensus on a tripartite process where luminal TGN protein sorting was also involved in segregating proteins prior to ISG formation, termed ‘*sorting by entry*’ [40]. Moreover, technological advances utilized by recent studies have revealed increasing levels of complexity, showing that some transmembrane components are in fact added to the SG after formation through retrograde plasma membrane/endosomal trafficking [46]. As we explore the luminal components of the insulin SG, we will come to appreciate that SG formation is difficult to lay out as a step-by-step mechanism. Individual components will contribute to multiple steps along the pathway, collaborating through a sequence of events to generate a functional entity that can be released upon stimulus. It is becoming more apparent that correctly forming this entity is crucial for systemic glucose homeostasis.

## 2. Luminal Components of the Insulin Secretory Granule

The luminal components of the insulin SG can be functionally segregated into four groups. These are cargo molecules, luminal enzymes and chaperones, ions (and their transporters and channels), and sorting receptors.

### 2.1. Cargo Molecules

The primary cargoes of SGs in the pancreatic β-cell are insulin, islet amyloid polypeptide (IAPP), the granins [chromogranin A (CgA), chromogranin B (CgB), secretogranin II (SgII), secretogranin III (SgIII), and VGF (non-acronymic)], and each of their precursors and derivatives. In addition to those covered in this review, the insulin SG also contains amines such as dopamine and serotonin [47,48,49], as well as nucleotides like ATP [50], which can be taken up by SG-localized pumps but as of yet, have ill-defined intragranular and post-exocytotic roles [51]. In this section, we will demonstrate what is known about the trafficking and processing of each individual cargo protein. These events are heavily dependent on the differential ionic composition of each compartment, where Ca^2+^, H^+^ and Zn^2+^ supplied by localized uptake pumps exist in an ascending concentration gradient proximal to distal (Figure 1).

*Insulin*. Insulin is synthesized as pre-proinsulin on the rough ER, and upon translocation has its N-terminal 24-residue signal sequence cleaved to form proinsulin [52]. Proinsulin undergoes folding in the ER where it acquires three disulfide bonds and dimerizes prior to ER exit [53,54]. En route to the TGN, proinsulin forms hexamers in the presence of Zn^2+^ [53,55], and importantly, proinsulin hexamers remain soluble [43]. Zn^2+^ binds to a histidine corresponding to residue 10 on the B chain of mature insulin (His-B10), and while the precise cisternal location of this event is undetermined, there is evidence of a Zn^2+^-dependent rate limiting step for proinsulin trafficking around the TGN/ISG compartment [54].

After entry into the ISG, proinsulin is converted to insulin and C-peptide via ordered cleavage at two sites of dibasic amino acid residues by the subtilisin-related proprotein convertases, first by PC1/3 and then by PC2 (Figure 2A). The 31–32 Arg-Arg site is located at the C-peptide/B-chain junction and the 64–65 Lys-Arg site is located between the C-peptide/A-chain junction. Molecular modelling suggests that the co-ordination of Zn^2+^ by His-B10 works to position these sites along the exposed radial surface of the proinsulin hexamer [56], enabling accessibility for the two processing enzymes. PC1/3 preferentially cleaves the B-chain junction on the carboxyl side of Arg32, generating a proinsulin intermediate split between residues 32 and 33 (*split 32,33* proinsulin) [4,5,57]. PC2 preferentially cleaves the A-chain junction on the carboxyl side of Arg65 to generate the *split 65,66* proinsulin intermediate [4,5,57]. Following conversion by each of the subtilisin-related prohormone convertases, the exoprotease carboxypeptidase H/E (CPE) acts to trim the revealed dibasic residues to create the ‘*des*’ intermediates, *des 31, 32*, or *des 64,65* proinsulin, with numbers denoting the excised residues [58]. A second round of endoprotease and CPE activity will generate insulin and C-peptide in a 1:1 molar ratio [3,4]. The insulin molecule consists of an A-chain and a B-chain, linked together by two disulfide bridges and maintained in hexameric oligomers through the co-ordination of two-Zn^2+^ by three of the six His-B10s [8,59]. Continual uptake of H^+^ and Zn^2+^ into the developing SG affects the charge state of hexameric insulin and facilitates its packing into extremely insoluble crystals [60]. The low percentage of unprocessed/incompletely processed proinsulin can pack with crystalline insulin to some extent [61] and C-peptide can co-precipitate with insulin in pH conditions mimicking the MSG [62]. C-peptide can also undergo further exoproteolytic cleavage to generate *des 27–31* C-peptide, accounting for roughly 10% of the total C-peptide content [63]. Upon exocytosis, exposure to the neutral extracellular pH is likely to dissipate the insulin crystal rapidly [64], allowing monomeric insulin to circulate and signal via the insulin receptor expressed on target tissues.

A human mutation of His-B10 to aspartate (mAsp-B10) underlies familial hyperproinsulinemia [65] and represents a condition where mutant proinsulin is presumed to be excluded from wild-type proinsulin hexamers. While expression of this mutant in mice does not affect its intracellular conversion to insulin, there is an enrichment of non-crystallized SGs, and the constitutive release of proinsulin is increased by ~15% [66]. These phenotypes could indicate that mAsp-B10 proinsulin is correctly targeted into the ISG, but there is an increased constitutive-like release in the absence of Zn^2+^-facilitated hexamarization prior to its conversion into insulin. Indeed, while contributing to the maturation of the SG, constitutive-like secretion is estimated to account for only 0.6% of the release of non-converted proinsulin [67]. This situation could represent an extreme example of protein exit out of the ISG, displaying the secretory capacity of the constitutive-like pathway. An alternative (and not mutually exclusive) explanation is that mAsp-B10 proinsulin leaks directly into the constitutive pathway from the TGN, however mAsp-B10 proinsulin degradation is also enhanced [66] suggesting that its transit to the PM occurs through the constitutive-like pathway (a route that travels via the endo-lysosomal system [43]). Nonetheless, these studies have highlighted that Zn^2+^-facilitated hexamarization is a primary mechanism of proinsulin sorting and consequent SG maturation.

Early studies investigating human proinsulin and rat proinsulin isomers I and II in primary islets revealed that they are differentially processed. Human proinsulin tends to be cleaved first at the B-chain junction to produce *des 31,32* proinsulin [68] (Figure 2C), whereas the rat isomers tend to be cleaved first at the A-chain junction to produce *des 64,65* proinsulin [69]. This is thought to be due to the amino acid located four residues prior to the cleavage site (P4 position [70]), where the presence of a basic lysine or arginine residue enhances substrate recognition and/or enzymatic activity [71]. Both rat isomers contain a basic arginine at P4 in the A-chain site, and both rat proinsulin I and human proinsulin contain a basic lysine at P4 in the B-chain site [72]. As a result, rat proinsulin I is more rapidly converted into insulin, and the accumulation of processing intermediates from this isomer is reduced due to the existence of basic residues at P4 in both cleavage sites [69].

Processing of human proinsulin follows a sequence favoring the prior activity of PC1/3 on the B-chain junction, followed by PC2, which has a far better affinity for *des 31,32* proinsulin than intact proinsulin [73]. Although this points to the existence of sequential cleavage through the action of both endoproteases, multiple lines of evidence indicate that PC1/3 works alone to produce mature insulin from both rat and human proinsulin isomers. While each enzyme possesses the catalytic ability to cleave at both dibasic sites [74], PC1/3 achieves this far more efficiently than PC2 [75,76,77] and processing intermediates of proinsulin accumulate when PC1/3 expression is low [76]. The situation is different in mice, seemingly requiring the activity of both endoproteases; while the deletion of PC1/3 from mice results in an extremely pronounced block in proinsulin conversion [78], knockout of PC2 also significantly hampers insulin maturation despite the presence of PC1/3 [79]. Finally, a recent study that re-characterized the expression of PC1/3 and PC2 in human islet β-cells found an abundance of PC1/3 and an absence of PC2, suggesting that PC1/3 is sufficient for humans to produce insulin [80]. Interestingly, humans with T2D had upregulated PC1/3 and an induction of PC2 expression. The authors of this study speculated that aberrant PC2 expression could cause a processing defect that underlies the pathological state, although, it may be the case that PC2 expression is invoked by metabolic stress as a compensatory response to assist PC1/3 in proteolytic activities. Indeed, the catalytic rate of PC2 on des 31,32 proinsulin exceeds that of PC1/3 on intact proinsulin [73]. Simple overexpression of either PC1/3 or PC2 has been shown to enhance proinsulin conversion in rat insulinoma INS1 cells [81], hence, induction of PC2 activity could support proinsulin conversion when PC1/3 is overwhelmed especially considering that the A-chain junction is not preferred by PC1/3 [4,57].

Both PC1/3 and PC2 endoprotease activities are sensitive to pH and Ca^2+^. In vitro assays using enzymes isolated from rat islets have shown that PC1/3 requires millimolar levels of Ca^2+^ and a pH close to 5.5 for activity whereas PC2 can exert activity at a micromolar levels of Ca^2+^ and over a broader pH range, although its pH optimum is also 5.5 [4]. In cells however, PC1/3 undergoes fast maturation into an active enzyme upon entry into the SG [82]. Due to the stringent regulation of PC2 by the molecular chaperone 7B2 [83,84,85], the low pH requirement for its autocatalytic activation [82,86], as well as its substrate-specificity to *des 31,32* proinsulin [73], its activity is likely to be restricted to later stages of SG maturation. Therefore, it appears that early PC1/3 activity at both sites could render PC2 redundant, as has been demonstrated in animal models [75,76,77] but not quite yet in humans. Crucially, compensatory upregulation of the endoproteases may be futile, considering the premature ISG release that occurs in β-cell failure. It has been known for some time that *des 31,32* intermediates are the predominant species of circulating proinsulin that is elevated in human T2D [87,88], therefore fast endoprotease activity is critical for systemic metabolic homeostasis. Therapeutic compounds that alter the ionic composition of the SG to bolster endoprotease activation and activity could be effective in treating T2D.

Despite a common outcome, nuances in the generation of insulin are clear between species. Their awareness may be important for translating data from model organisms to the context of human β-cell function.

*Islet Amyloid Polypeptide.* IAPP is a 37 amino-acid peptide stored in the MSG that is co-secreted with insulin in a 1:100 molar ratio [89,90,91], and can function to suppress insulin secretion and control various aspects of energy homeostasis [22,92]. Additionally, known as amylin, IAPP and its precursors and derivatives are notorious for forming fibrils that distribute extracellularly throughout islets as amyloid deposits, a pathological feature of human T2D [93]. Early observations report the occurrence of islet amyloid deposits in >90% of diabetic patients [94,95] but later studies have shown a variable prevalence depending on duration of disease and ethnicity, especially when sample size is increased [96]. The question of how IAPP remains non-pathogenic in healthy conditions and how it transitions to a pathogenic molecule has kept researchers occupied for some time. Appropriately, most work on IAPP has been focused on its secretory dynamics, processing, and amyloidogenic properties [97,98,99] rather than its luminal sorting. To this end, not much is known about its behavior in the early secretory pathway or the determinants of its trafficking fate.

Akin to proinsulin, human pro-IAPP is a 67 amino-acid (aa) peptide derived from pre-proIAPP that forms an intramolecular disulfide bridge in the ER and is subject to endoproteolytic processing [100,101,102,103] (Figure 2B). It is thought that PC1/3 acts first on a C-terminal proregion in the TGN [102] which is followed by CPE action to generate a 48-residue processing intermediate. A subsequent round of PC1/3 or PC2 and CPE action on the N-terminal proregion generates a 38 aa peptide with a C-terminal Gly termed amylin free acid [104]. A fourth enzyme, peptidyl-glycine alpha-amidating monooxygenase (PAM), is probably responsible for amidation at the C-terminal 38 glycine residue which may or may not occur prior to cleavage of the N-terminal pro-region, to generate the mature C-terminally amidated form of IAPP [104]. Finally, a fifth membrane-bound enzyme that localizes to the β-cell SG, β-secretase 2, can process IAPP further into smaller fragments [105].

Early reports demonstrated that PC2 can cleave at both the N- and C-terminal proregions of pro-mouse IAPP (mIAPP) in addition to PC1/3 cleavage occurring only at the C-terminal site [100,101,102,103]. However, both rat and human islets appear not to express PC2 at detectable levels normally [80]. Indeed, pro-human IAPP (hIAPP) can be fully processed by PC1/3 in PC2 null mice [106]; however, pro-rat IAPP (rIAPP) can be processed at both sites by PC2 but only at one site by PC1/3 [103]. These differences are likely due to the modified sequence at the C-terminal site (Figure 3), where the position of Ala and Val residues may determine whether PC1/3 can cleave: i.e., KR↓VA (Val at P1′ and Ala at P2′) in rIAPP compared to KR↓AV (Ala at P1′ and Val at P2′) in hIAPP and KR↓AA (Ala at P1′ and P2′) in mIAPP. Likewise, it has also been suggested that a Val common to both hIAPP and rIAPP at the N-terminal site allows cleavage by PC1/3 [80], which is not experienced by mIAPP [100] that contains Met at this residue. Thus, PC1/3 may be sufficient for full pro-IAPP processing in humans due to the sequence variations that lie between species.

IAPP resides in the soluble fraction of the MSG, and hIAPP is extremely fibrillogenic whereas rIAPP and mIAPP are not fibrillogenic at all [107]. It has been shown in vitro that pro-hIAPP products become increasingly more amyloidogenic with further cleavages [108], however, it is thought that the presence of ionic and molecular factors in the MSG periphery inhibits hIAPP oligomer formation in healthy conditions [109,110,111,112,113,114]. An extensive body of literature has covered the molecular mechanisms of hIAPP pathogenicity in T2D [97,98,99]. Considering that hIAPP is likely not a driver of SG biogenesis or function but instead plays the role of chaotic passenger, we will focus here on how secretory pathway dysfunction could precede hIAPP-mediated β-cell damage.

A current working hypothesis to explain the initiation of islet amyloid formation is that hIAPP-related peptides undergo dysregulated fibrillogenesis at some point inside the β-cell secretory pathway, potentially due to overproduction during compensation/failure [115], but this only occurs within a small subset of islet β-cells [97,98,116,117,118]. If pro-hIAPP overproduction is the fuel, then disruption of organellar membranes from within the cell is the spark that results in β-cell death and the deposition of extracellular amyloid. Through regular exocytosis from neighboring β-cells, released hIAPP can then add to the size of the initial deposit. An alternative (and not mutually exclusive) hypothesis suggests that release of the 48-residue pro-hIAPP (pro-hIAPP_1–48_) intermediate can initiate extracellular amyloid formation by a specific interaction with heparin sulfate proteoglycans in the extracellular matrix [119]. Interestingly, large ordered fibrils that make up the bulk of the visually identified IAPP deposition are thought to be relatively inert on a cytotoxic level [99], although interruption of islet cytoarchitecture could impair coordinated islet function. Rather, it is the presence of medium-sized disordered oligomers that are thought to exert most of the cellular damage [120]. Fitting with this, recent experimental focus has instead been placed on the mechanism of medium-sized oligomer formation and cytotoxicity [121,122].

Not surprisingly, the N-terminal prosequence of hIAPP is detectable in islet amyloid deposits [123,124]. This observation resembles what is observed with proinsulin in that incompletely processed hIAPP may be released from the β-cell during T2D, and indeed, elevated serum pro-hIAPP has been observed in glucose intolerant and T2D patients [125]. If one considers that ISGs released during β-cell failure are a source of hIAPP processing intermediates (Figure 2C), we could look to luminal factors that might explain the propensity of these molecules to become pathogenic. Indeed, insulin, Zn^2+^, H^+^, Ca^2+^, C-peptide, and proinsulin have been assessed individually or in combination, in vitro, along with hIAPP, in various studies to reason that a delicate balance of cofactors is required to inhibit hIAPP oligomerization [109,110,111,112,113,114,126]. In healthy cells, regulated exocytosis of MSGs could maintain this balance as components are released in an appropriate molar ratio. Conversely, during β-cell compensation and failure, release of the incompletely formed ISG might not replicate this outcome, and cytotoxic hIAPP oligomers could form in the extracellular microenvironment adjacent to the plasma membrane to induce membrane damage.

Dysregulated hIAPP oligomerization exacerbates the progression of T2D, so preventing β-cell death at the hands of hIAPP could limit T2D severity. Abnormal SG composition or the premature release of ISGs may be contributing factors, highlighting the importance of correctly forming the insulin SG.

*Granins.* The granin family of proteins (CgA, CgB, SgII, SgIII, and VGF) are ubiquitously expressed across neurons and endocrine cells and are considered major contributors to the biogenesis of regulated SGs from within the lumen (Figure 4). Their effectiveness has been displayed by several groups with findings that expression of just a single granin in cells that do not have a regulated secretory pathway is able to produce SGs that are capable of regulated release [24,127,128,129,130]. Granins are synthesized as soluble cargo precursor proteins, which are highly acidic and hydrophilic, but are prone to aggregation under mild acidity (pH < 6.4) and high Ca^2+^ (>1 mM) conditions [25,131]. It has been shown that both of these ionic requirements must be met for granin aggregation [131,132], which can be achieved at the initiating site of SG biogenesis in the TGN (Figure 1) where ion pumps maintain a high luminal Ca^2+^ concentration and contribute to a substantial lowering of the pH through enhanced H^+^ uptake [27,133,134,135]. Moreover, several granins have been shown to interact with each other, and, lacking transmembrane domains themselves, some can also interact with lipid species on the luminal aspect of the secretory pathway membrane to provide a link between soluble and membrane fractions. In this way, their physical abundance, coordinated aggregation within the TGN, and binding to specific components on the membrane has been proposed to drive the segregation and sorting of peptide hormones and other proteins into the regulated secretory pathway [23], meeting the requirements of a ‘refined bulk-flow’ *sorting by entry* mechanism [40].

Sphingolipid–cholesterol lipid rafts accumulate in microdomains of the TGN [136,137], and these rafts can alter the distribution of transmembrane components to create sorting stations that are essential for granule biogenesis [138,139]. These rafts are also enriched in vesicles of the regulated secretory pathway [140,141,142,143], indicating that SG membranes originate from the sorting domains where granins and other SG constituents aggregate and bind. Importantly, saturated fatty acid and cholesterol intake can change the composition of lipid species distributed among cell membranes to influence trafficking and SG morphology [144,145]. Therefore, dietary status could affect interactions between the granins and secretory pathway membranes, but this requires investigation. Finally, since granins can bind Ca^2+^ at a high capacity with low affinity [132], they are also thought to equip the SG with the ability to store and release Ca^2+^ [146,147].

It should be noted that most of the literature on granins has reported on their role in neuronal/neuroendocrine cell lines, which although share features in common with the β-cell, have secretory pathways adapted to the specific needs of neural transmission. In these settings, we can draw insight from molecular interactions that govern trafficking and behavior of the granins themselves, but specific effects of granin depletion on SG biogenesis/secretion are often subject to cell-specific variation and thus will only be discussed with respect to the β-cell.

Granins can possess multiple sorting determinants and may be targeted to several SG sub-populations (Figure 4). SgIII is membrane-associated, and contains an N-terminal lipid-binding region that is required for its sorting into SGs of AtT-20 cells [148] and for its interaction with cholesterol in INS1 and AtT-20 SGs [149]. This suggests that SgIII is sorted into the regulated secretory pathway through an interaction with TGN cholesterol [148]. N-terminal residues (48–111) of CgA can bind to SgIII to follow SgIII sorting into the regulated secretory pathway of AtT-20 cells, where it also exists in association with SG membranes [149]. Importantly, CgA also associates with INS1 granule membranes but only in the presence of SgIII [148]. These results collectively indicate that SgIII is an adaptor for CgA in β-cells, with both granins associated at least to some degree with the SG membrane, and that correct trafficking relies on the presence of an N-terminal region on SgIII that binds cholesterol. This has been further demonstrated in PC12 cells, where SgIII was specifically shown to sort large aggregates of CgA into SGs [150]. Very little is known about the trafficking determinants of SgII aside from an understanding that both the N- and C-termini contain information for sorting [151], and that it may interact with SgIII on the SG membrane [152,153]. While SgII also regulates granule biogenesis in other secretory cells [154], interactions have not been published in β-cell models, and in general, insight into SgII function in the β-cell is lacking.

Like SgIII, CgB can interact with cellular membranes via a highly conserved N-terminal 22 residue disulfide-bonded loop [155,156]. This loop is essential for CgB sorting to SGs but is not required for its aggregation within the TGN, indicating that CgB aggregates are not routed by default to SGs but are sorted through mediation of exposed N-terminal loops with the TGN membrane [155]; although, the corresponding membrane component is yet to be found. Several observations suggest that CgB trafficking is not entirely synchronous with insulin. In addition to the insulin SG, CgB also occupies distinct granules that do not contain insulin and conversely, insulin can occupy SGs devoid of CgB [157]. Additionally, CgB is present with SgII in nucleoplasmic vesicles of bovine chromaffin cells where they may have a role in regulating nuclear Ca^2+^ homeostasis [158], although this has not been studied in the β-cell. In the β-cell, CgB co-localizes and co-immunoprecipitates with VGF [159], and it has been shown in vitro that CgA and CgB can form dimers at pH 7.5 and heterotetramers at pH 5.5 [160], suggesting that CgB could traffic with either VGF or CgA. However, VGF does not immunoprecipitate with CgA [159]. Little else is known about the determinants of VGF trafficking, although a predicted alpha helix loop in its C-terminus may be required for direction into INS1 SGs [161].

A handful of studies have investigated the consequences of granin depletion in β-cells albeit with varying success, possibly due to the method of study. Transient gene silencing seems to outcompete stable knockouts for studying function, and this is probably due to the circumvention of compensatory changes that occur during development. For example, whole body CgB knockout (KO) provided an insulin secretory defect that was unable to be explained aside from a small decrease in the number of docked SGs [162], whereas adenoviral knockdown (KD) of CgB in INS1-832/3 insulinoma cells and isolated mouse islets resulted in marked insulin secretory defects that could be explained by a defect to SG biogenesis [159]. Similarly, islets from whole body CgA KOs actually have enhanced insulin secretion with no defects to SG generation [163], whereas siRNA KD of CgA in the human β-cell line, EndoC-βH1, resulted in reduced basal and glucose-stimulated insulin secretion (GSIS) as well as cellular insulin content [164]. CgA KO mice had compensatory doubling in CgB expression and tripling in SgII expression [163], which may explain the absence of a secretory phenotype in CgA KOs. Islets from whole-body SgIII KO mice have impaired GSIS but only when subject to a high-fat-high-sucrose diet. This is associated with reduced insulin and increased proinsulin content, but there were no reported ultrastructural granule abnormalities [165]. Interestingly, in this study, CgA levels failed to increase when SgIII KO mice were put on diet but did so in the islets of wild-type mice [165]. As discussed previously, SgIII is a known adaptor for CgA and therefore its absence could result in CgA mis-sorting and thus the failure of compensatory upregulation. Finally, VGF depletion has also been assessed via KD of its mRNA in INS1-832/3 cells and a tamoxifen-inducible KO from mouse islets [166]. This study noted reduced GSIS in both models, associated with a loss of total and docked SGs, and a reduction to their size in line with an increased cellular proinsulin-to-insulin ratio and delays to proinsulin conversion [166]. This study concluded that VGF depletion caused a granule replenishment defect, hampering the secretion of newly synthesized granules during the sustained second phase of GSIS [166].

In summary, the granins are critical components of ISG formation, driving the formation of regulated carriers from within the lumen through aggregating and binding to distinct sites of the TGN membrane. Their ubiquity across cells of the neuroendocrine system implies an essential role for SG function, where their combined abundance and aggregative nature may confer unique characteristics to the SG.

### 2.2. Luminal Enzymes and Chaperones

Several enzymes and chaperones undergo processing in the secretory pathway and are targeted to the β-cell SG to generate the diverse intragranular cocktail. PC1/3, PC2, CPE, the PC2 binding partner 7B2, PAM, furin, chaperonin 60, and β-secretase 2 are members of this functional group that have activity in the β-cell. Active furin is widespread across the trans Golgi network, cell surface and endosomes, however it traverses the ISG before being sorted out of the maturing granule [167]. β-secretase 2 is a transmembrane aspartic protease that was previously mentioned for its role in mature IAPP proteolysis [105,168]. Chaperonin 60, a heat shock related protein, has also been found to co-localize and co-immunoprecipitate with proinsulin and PC1/3 [169]; however, the functional significance of this protein has not been investigated. Since we have already discussed the activity of CPE, PC1/3 and PC2 on proinsulin and proIAPP conversion, here we will restrict their discussion to trafficking and activation.

*Carboxypeptidase E.* CPE, also known as CPH, exists in both soluble and membrane-associated forms in β-cells [170]. An alpha-helix in the C-terminus of CPE anchors through cholesterol rich lipid rafts of the secretory pathway membrane, leaving six residues protruding to the cytoplasm [171,172] (Figure 5). Importantly, penetration through the membrane only occurs at or below pH 6 [172], conditions reflecting the late Golgi and SG compartments but not the proximal Golgi or ER [173] (Figure 1). Several lines of evidence have also demonstrated that membrane-bound forms of CPE can aggregate at this pH with at least 1 mM Ca^2+^ [174], and co-immunoprecipitate in these conditions with both pro-opiomelanocortin and insulin in vitro [175]. This aggregation appears to occur independently from membrane binding, since treatment with Triton X-100 at pH < 6 to interfere with aggregation does not dissociate CPE from the membrane [174]. Collectively, pH-dependent lipid-raft insertion and aggregation provide a means by which CPE is concentrated along the TGN membrane for targeting to the SG. Moreover, it has been shown that CPE interacts with SgIII in both INS1 and AtT-20 cells [176], providing more control over targeting. Following SG entry, CPE is cleaved by an endoprotease at its C-terminus to generate the soluble, major enzymatic form of CPE [177] (Figure 5). Immunostaining reveals its predominant localization to the SG at steady state [178], where its enzymatic activity operates in a narrow pH optimum between 5.0 and 5.5 [179].

*PC1/3*. The trafficking and activation of PC1/3 is considerably less complicated than PC2. In the ER, pre-proPC1/3 undergoes cotranslational signal peptide removal to generate a 94 kDa pro-form of PC1/3 [180,181]. This precursor harbors enzymatic activity but only toward its own N-terminal pro-region, which is cleaved in the ER [180,181] and thought to remain associated with the catalytic site of mature 87 kDa PC1/3 to prevent activity in the early secretory pathway [182] (Figure 5). This will later dissociate after a second cleavage by PC1/3 [182]; however, the location of this event (TGN or ISG) is not settled yet. Moreover, the C-terminal region partially inhibits PC1/3 activity [183,184], and befittingly, PC1/3 can process certain substrates prior to its entry and complete activation inside the SG [185,186]. The C-terminal region also contains a predicted alpha helix required for sorting into the regulated secretory pathway [187], likely through an interaction with membrane lipid rafts [188]. Following entry into the SG, the inhibitory C-terminal region is cleaved (possibly by itself) to generate fully active 74 and 66 kDa products [183,184,189] (Figure 5), providing a relatively simple activation mechanism in the SG. 87 kDa PC1/3 exhibits a pH optimum between 5.5 and 6.5 [190], respective conditions reflecting the SG and the TGN (Figure 1). Both 74 and 66 kDa products exhibit pH optima between 5.0 and 5.5 [4,189], reflecting the MSG. All forms also have a high Ca^2+^-dependence [189,190], so the luminal ionic composition must be optimized for PC1/3 activity.

*PC2 and 7B2*. In the ER, pre-proPC2 undergoes cotranslational signal peptide cleavage to generate a 76 kDa proPC2. Unlike proPC1/3, its inhibitory N-terminal pro-region is not cleaved and remains associated with the catalytic subunit until it reaches the SG [191]. ProPC2 traffics through the secretory pathway together with its chaperon 7B2 [192]; after synthesis and folding, proPC2 can bind pro7B2 in the ER where it accelerates proPC2 trafficking to the Golgi [83,193,194]. Pro7B2 is proteolytically cleaved at Arg152 (a furin cleavage site) in the TGN to generate a 21 kDa N-terminal (7B2-NT) and a 5 kDa C-terminal peptide (7B2-CT) [85,195], both of which remain associated with proPC2 [86]. While the 7B2-NT appears to maintain proPC2 folding and trafficking, 7B2-CT is a well-established PC2 inhibitor in vitro [196] (Figure 5).

Both proPC2 and 7B2-NT can aggregate under pH- and Ca^2+^- conditions mimicking the TGN [197,198]. Residues 45–84 in proPC2 have been shown to mediate its association with TGN membranes [199]. Here, proPC2 likely interacts with sphingolipids in the TGN membrane since sphingolipid depletion causes re-routing of transfected mature PC2 to the constitutive pathway [199]. ProPC2 also requires 7B2 for direction [200], so 7B2 depletion will cause proPC2 to traffic constitutively [201]. Therefore, in the absence of 7B2-peptides, unfolded, improperly aggregated PC2 could route constitutively [192,201]. In the SG, proPC2 does not undergo full autocatalytic maturation until the luminal pH drops to 5.2 [86,191], conditions reflecting the MSG (Figure 1), and once fully mature PC2 cleaves and removes the inhibitory 7B2-CT fragment [84]. Thus, although PC2 is active in vitro over a broad range of pHs and Ca^2+^ [4], its fully active form is restricted to the MSG within cells.

*PAM*. PAM is a bifunctional enzyme consisting of two contiguous catalytic domains, peptidylglycine alpha-hydroxylating monooxygenase (PHM) and peptidyl-ɑ-hydroxyglycine ɑ-amidating lyase (PAL) [202] (Figure 5). Sequential action of PHM followed by PAL functions to amidate glycine at the carboxyl terminus of peptides [203], which removes charge to confer full biological activity to the peptide [204]. Cargoes that have already been processed by PC1/3, PC2 and CPE to yield C-terminal glycine residues are generally subject to this reaction [205]. While PHM is active over a broad acidity range [206], the stability of the intermediate formed by PHM declines at pH levels >6.0 [207] and the pH optimum for PAL is around 5.0. At least 50% of all biologically active peptides require amidation for full biological activity [208], and thus far, PAM is the only enzyme identified to be responsible for this reaction in vivo. In both human and mouse islets, PAM co-localizes with insulin, glucagon, and somatostatin [164,209], and while insulin is not a target of PAM, IAPP is a potential target [104] and CgA was recently verified as a PAM substrate in β-cells [164]. PAM depletion can affect insulin content and GSIS which may be mediated by its downstream targets including CgA [164]; however, PAM haploinsufficiency in mice does not accelerate diet- or IAPP-induced diabetes [209]. Notably, T2D-associated PAM risk alleles exist that are associated with reduced insulinogenic index [164,209], thus, PAM activity appears to have relevance to β-cell function.

Several isoforms of PAM differentially expressed between tissues [210] are produced by alternative mRNA splicing; some are soluble and others are type-I transmembrane enzymes [211]. Human β-cells only express soluble isoforms whereas mouse β-cells only express transmembrane isoforms [209]. Isoform differences are due to the presence/absence of specific regions, including a C-terminal transmembrane region, a linker region between the two contiguous enzymes PHM and PAL, as well as each of the two enzymes themselves [210]. Both forms are enzymatically active although integral membrane PAM will presumably have more restricted access to substrates [212].

Both soluble and integral forms of PAM traffic simultaneously through the early secretory pathway, but then diverge upon entry into the ISG [213]. The enzymatic domains of PAM contain information for direction toward the regulated secretory pathway since expressed soluble PHM or PAL traffic correctly into ISGs [214], however, the transmembrane/cytosolic aspect of integral membrane PAM can override luminal trafficking information for transit via an independent route [213,215]. While both forms enter the ISG efficiently and are retained to some degree within maturing SGs, they can both exit the ISG through the constitutive-like pathway in unstimulated conditions [213]. Integral membrane PAM exits the SG to a greater extent [213], and has been shown to cycle through the PM where it may be retrieved to the TGN [216] or the MSG [217,218].

Since human β-cells only express PAM3 [209], a soluble isoform, its trafficking is relatively simple. The situation is more complicated in mouse β-cells which express transmembrane isoforms PAM1 and PAM2 [209], and thus, are subject to additional trafficking fates and require endoproteolytic cleavage within the SG to generate soluble PAM to access substrates more readily [212]. In addition to a low pH [206,212], Zn^2+^ and Ca^2+^ [219], PAM requires Cu^2+^ [220] and ascorbate [203] for activity.

Collectively, enzymes control the intragranular landscape by modifying proteins and altering their properties, operating as the focal regulatory units of the SG lumen. This culminates in the main transformative event of the granule interior—the crystallization of proteolytically generated insulin—which creates an extremely dense proteinaceous core. Though deploying enzymes that require such specific conditions for activity, it appears that the β-cell strikes a balance between rapid processing and orderly aggregation to ensure the safe generation of a functional product.

### 2.3. Ions, Transporters, and Channels

Transporters and channels embedded into the secretory pathway membrane control the composition of the luminal milieu to facilitate cargo sorting and processing. They also control the release of ions from the SG store to regulate cytosolic events. In this section, we will discuss the coordinated function of transporters and channels that modulate important features of the intergranular lumen.

*Activation: H^+^.* Foundational studies from the Hutton lab established the low pH of the β-cell SG [221] and found activity of an ATP-dependent pump responsible for translocating H^+^ into the granule lumen [26]. The vacuolar H^+^-ATPase (V-ATPase) localizes to the β-cell SG [133] and is the major component responsible for endoprotease activation and cargo processing in regulated secretory cells [27,134]. Due to the influx of positive charge, a complimentary influx or efflux of anions or cations, respectively, would be required to maintain a normal electrochemical gradient across the SG membrane and this is normally achieved by the counterregulatory actions of Cl- transporters [222,223,224,225,226]. Moreover, other transporters can utilize the proton gradient generated by V-ATPase to exchange H^+^ for cytosolic materials. For example, vesicular monoamine transporter type-2 (VMAT2) can exchange intragranular H^+^ for cytosolic monoamines, thus functioning to regulate the luminal pH [227].

V-ATPase is widely distributed throughout the endo-lysosomal system and the plasma membrane. It consists of two complexes, V_1_ (cytosolic) and V_0_ (membrane-associated) (Figure 6), which, respectively, contain eight and nine subunits [228]. ATP hydrolysis by V_1_ provides energy for V_0_ to rotate and pump 2–4 H^+^ molecules from the cytosol into luminal or extracellular spaces [229], and this can be controlled by several factors. Dissociation of V_1_ from V_0_ is the primary mechanism of pump regulation [230] and is sensitive to glucose exposure [231]. Complex assembly is affected by the membrane lipid composition; the β-cell SG contains an abundance of enriched lipid species, and sphingolipids are thought to stabilize assembly and facilitate ATP hydrolysis [232]. Localization and density of the pump is obviously limiting for compartmental acidification, thus, regarding the β-cell SG, an abundance of V-ATPase is critical for luminal protein processing.

In addition, H^+^ pumping is sub-optimally coupled to ATP hydrolysis [233,234], providing room for further V-ATPase modulation. This may contribute to the establishment of a secretory pathway H^+^ gradient or allow V-ATPase to respond to environmental stimuli. In yeast, V-ATPase efficiency is associated with the *a*-subunit of the V_0_ domain, which is also likely to determine its cellular localization [235]. Subunit *a* is situated in an ideal position to modulate the pump. It is embedded into the membrane adjacent to a proteolipid ring formed by the *c* subunits of V_0_ (through which H^+^ passes) and extends toward the cytosol to interact with V_1_ [228]. Yeast co-express two homologs to the mammalian *a*-subunit and these have been shown to affect both the localization and activity of V-ATPase [235].

Four isoforms of subunit *a* exist in mammals (*a*1–*a*4) making it the most diverse member of the V-ATPase complex, suggesting that this component can endow compartmental specificity regarding the localization and efficiency of the pump. *a*1 appears to localize to the Golgi, *a*2 to both lysosomes and the Golgi and *a*3 is mostly expressed on β-cell SGs [133]. Interestingly, the *a*4 subunit has been shown to interact with 1-phosphofructokinase (PFK1) in the human kidney [236], coupling nutrient sensing to V-ATPase activity [237]. SGs with low pH are required for GSIS [238]; therefore, glucose-mediated acidification could facilitate both SG maturation and release.

Accessory subunits Ac45 and the prorenin receptor, encoded by *ATP6ap1* and *ATP6ap2* genes, respectively, are also associated with the V-ATPase to assist SG acidification in β-cells [239,240,241,242]. Ac45 is subject to processing by furin [239,242] and the prorenin receptor interacts with and may act downstream of the GLP1 receptor [240,241]. A V-ATPase interactor, Wolfram syndrome 1 (WFS1), resides in β-cell ER and SG membranes [243] and has been shown to bind the V_1_A subunit via an N-terminal region to assist SG acidification [244]. To conclude, the V-ATPase and/or its regulatory subunits are potential targets for enhancing insulin production during high demand.

*Crystallization: Zn^2+^.* Insulin SGs hold high levels of Zn^2+^, with some estimates approaching 30 mM [29,245]. Zn^2+^ can alter the structure of its bound proteins, cofactor for enzyme activity and also serve as an extracellular signaling molecule [246]. Measurement of insulin and Zn^2+^ released during GSIS reveals that the total amount of Zn^2+^ in the SG is at least double that which is expected based on the stoichiometric composition of the insulin hexamer [247], probably due to the existence of an additional Zn^2+^ that displaces water within crystallized hexamers [28]. Co-secreted Zn^2+^ has autocrine [248,249] and potentially paracrine [250,251] effects on other islet cells, and may also travel to the liver to inhibit insulin receptor endocytosis and thus hepatic insulin uptake [252]. One study using computer modelling has also suggested that Zn^2+^ could maintain insulin in an oligomeric state during secretion, and that this would limit the availability of Zn^2+^ and monomeric insulin to act as anti-fibrillogenic agents against hIAPP-related proteins [253]. Early estimates, however, report that exposure to the extracellular environment would likely dissipate the insulin-zinc hexamer seconds after complex dilution outside of the β-cell [64], allowing monomeric insulin to circulate and signal. Notably, reductions in circulating and pancreatic Zn^2+^ levels are implicated in those with excess fat and T2D [254,255], and Zn^2+^ supplementation can enhance insulin secretion to better control glycemia during the insulin resistant state [256]. Thus, intracellular changes in Zn^2+^ could drive aspects of β-cell dysfunction.

Secretory pathway Zn^2+^ uptake is under the control of the Zn^2+^ transporter (ZnT) family. ZnTs dimerize to localize and function and likely do so as H^+^/Zn^2+^ antiporters [246,257], so therefore the establishment of a luminal proton gradient by V-ATPase may be permissive for Zn^2+^ uptake. ZnT5 and Znt7 localize to the β-cell Golgi apparatus whereas ZnT5 and ZnT8 are in the SG [28] (Figure 6). ZnT3 has been shown to colocalize with insulin in INS1 cells [258], however it appears to be absent in β-cells from mouse islets [259]. ZnT8 is the most abundantly expressed β-cell ZnT but has minimal expression in other tissues [28].

The current literature on ZnT8-mediated Zn^2+^ homeostasis is deep and interesting, owing to the existence of multiple T2D susceptibility loci covering the *SLC30A8* gene [260,261,262]. Importantly, loss of function mutations at residue 325 tend to favor a reduced risk of T2D, and opposite deleterious effects are seen with a gain of function at this residue. These may be mediated by changes to proinsulin conversion and insulin secretion, raising the question of whether Zn^2+^ is important for β-cell SG biogenesis. Surprisingly, guinea pigs express a proinsulin molecule that lacks histidine at the B10 residue and thus cannot bind Zn^2+^, similar to the human mutation discussed in Section 1, yet they can generate SGs, process proinsulin, and secrete insulin [263]. However, despite the inability of insulin to bind Zn^2+^, the presence of Zn^2+^ in these settings is not changed as is the case with altered ZnT8 transporter activity.

Studies of the effect of ZnT8 depletion in β-cell lines and rodent models have provided results that are difficult to synthesize thus far [261,262,264]. In general, experimental depletion of ZnT8 does not lead to major impairments in proinsulin processing or insulin content; however, SGs tend to be void of electron-dense spheres, appearing instead as electron-dense rods or as pale ISGs. This agrees with the expression of other ZnT members through the secretory pathway and on the SG, providing sufficient Zn^2+^ to sustain intragranular functions albeit with impaired insulin crystallization [28,265]. Assessment of insulin secretion in various ZnT8-deficient models has shown a mixed bag of results, with reports contrasting between mildly reduced, unchanged, or mildly enhanced effects [261,262,264]. Recognition of variable factors such as the mode of deletion, and mouse age, sex, and genetic background are hoped to assist in study design to clear up the matter [261]. Indeed, a study published in 2020 revealed both positive and negative age-dependent consequences of ZnT8 deletion in mice [266]. Moreover, assessing the importance of Zn^2+^ for SG biogenesis is complicated by the presence of multiple ZnTs. For example, an investigation comparing the effects of a whole-body single ZnT7 KO to a double ZnT7/ZnT8 KO, revealed that the double KO could provide a significant secretory defect not seen by ZnT7 KOs alone or by previous reports in ZnT8 KOs [267].

Several lines of evidence indicate that ZnT8-mediated Zn^2+^ import affects proinsulin processing, which could explain, at least in part, the protective or deleterious impacts of *SLC30A8* mutations. It has been shown that humans with the arginine risk variant of ZnT8 at residue 325, thought to be a gain-of-function mutation [268], display increased circulating proinsulin compared to the protective tryptophan variant [269]. Moreover, mice overexpressing the arginine 325 variant have an increased β-cell proinsulin content and release [270,271]. Conversely, in INS1E cells expressing the loss-of-function tryptophan 325 variant, cyclosporin A-induced β-cell dysfunction was attenuated when compared to the gain-of-function arginine 325 variant [272]. Finally, humans heterozygous for a ZnT8 variant truncated at residue 138, which impairs transporter synthesis and thus results in ZnT8 haploinsufficiency, are equipped with increased proinsulin conversion and circulating insulin, and improved glucose tolerance [270,271]. Since ZnTs are likely to function as H^+^/Zn^2+^ exchangers [246,257], enhanced flux through ZnT8 could buffer the reduction in luminal pH during SG maturation to limit endoprotease activity. Accelerated packing of the granule matrix in the presence of high Zn^2+^ could also reduce the accessibility of proinsulin to the endoproteases for conversion. In the case of reduced ZnT8 activity or haploinsufficiency, the presence of adequate Zn^2+^ supplied by the single copy of ZnT8 or other ZnTs in the secretory pathway would probably maintain hexameric proinsulin transit and cofactor for CPE activity, while the event of insulin crystallization does not appear to be necessary for regulated release. Therefore, in the setting of β-cell compensation, a ready supply of mature insulin could help to control circulating glucose. Importantly, *SLC30A8* risk-alleles are particularly noted to confer T2D susceptibility to lean individuals [262,273], so in this case, limited proinsulin conversion in newly synthesized granules that are preferentially secreted could reduce the insulin response to glucose. Nonetheless, there is still plenty to be learned about the impact that Zn^2+^ has inside the insulin SG. While the consequence of *SLC30A8* gene variants have been of interest, little is known about Znt8 structure and mechanism, and the factors that regulate it.

*Modulation: Ca^2+^.* Ca^2+^ is concentrated within the secretory pathway relative to the cytosol, where it controls luminal protein sorting and processing among other activities. Travelling proximal to distal, the amount of free Ca^2+^ decreases whereas the total amount of Ca^2+^ increases (Figure 1), in line with several proteins harboring an increased affinity to Ca^2+^ when the pH is reduced [132,274]. Quantitatively, the β-cell SG can hold between 50 to 100 mM Ca^2+^ [29], although measurement of the free concentration yields values of around 50 µM (~0.05% of the total amount) [31]. Therefore, the SG compartment, endowed with transporters and channels for Ca^2+^, should possess a high capacity to buffer cytosolic Ca^2+^. This dynamic reservoir can be utilized during β-cell stimulation to sequester and release Ca^2+^, modulating the cytosolic signals that underlie insulin secretion. Indeed, it has been shown that depletion of Ca^2+^ from the SG compartment impairs exocytosis [275]. β-cell SGs have been shown to take up Ca^2+^ during nutrient exposure [31], and Ca^2+^ release from the SG is required for insulin secretion [276]. Ca^2+^-dependent effector proteins are in proximity to the SG and sites of exocytosis [37], and therefore localized Ca^2+^ released from the SG could facilitate its own trafficking and exocytosis.

Several transporters and channels act in concert to coordinate luminal and cytosolic events that are regulated by Ca^2+^ (Figure 6). The secretory pathway Ca^2+^ ATPase 1 (SPCA1) is an uptake pump that regulates a Ca^2+^-dependent secretory protein sorting mechanism at the TGN in constitutively secreting cells [277,278]. In INS1 cells, SPCA1 may sequester cytosolic Ca^2+^ into secretory pathway compartments during glucose stimulation, accounting for around 20% of the total SG Ca^2+^ uptake [279]. Its depletion thus enhances insulin secretion [279]. In these cells, SPCA1 was shown to fractionate strongly with insulin suggesting its localization throughout the early and late secretory pathway [279].

Ryanodine receptors (RyRs) respond directly to cytosolic Ca^2+^ to release Ca^2+^ from intracellular stores but are inactive at low and high concentrations of Ca^2+^. RyR-1 was identified as the lone RyR subtype expressed on the β-cell SG, but it is also expressed on the ER where RyR-2 also locates [276]. Pharmacological inhibition of SG RyR-1 reduces Ca^2+^ efflux and impairs GSIS [276].

NAADP is a potent signaling molecule produced during β-cell glucose metabolism [280] that binds to unidentified NAADP-sensitive sites to elicit Ca^2+^ release from intracellular stores, including the SG, during GSIS [31,281]. SIDT2 is located on insulin SGs and may mediate this mechanism [281]. Whole-body SIDT2 KO mice are glucose intolerant and have an insulin secretory defect [282]. The requirement of either NAADP or RyR-1-mediated Ca^2+^ release for GSIS suggests that the SG is an important reservoir of Ca^2+^ that is utilized during secretory functions.

After initial confusion [283,284], it was recently verified that all three subtypes of the IP_3_R are expressed on β-cell SGs at a level two-fold higher than the ER [30]. IP_3_Rs require tetramerization and binding of each member to inositol 1,4,5-triphosphate (IP_3_) in order for the channel to open and release Ca^2+^ from stores, and these tetramers can be formed by any combination of IP_3_R subtypes [285]. In the absence of IP_3_, Ca^2+^ inhibits the IP_3_R, although, Ca^2+^ must bind to the IP_3_R with IP_3_ for the channel to open [286]. IP_3_Rs localized on non-β-cell SGs have been estimated to be more sensitive than those of the ER [287], although one study reported that β-cell SGs do not release Ca^2+^ in response to IP_3_ [31]. Several caveats are apparent in this study; low levels of Ca^2+^ were incubated with permeabilized mouse insulinoma MIN6 cells exposed to IP_3_ despite a high requirement for maximal IP_3_R activation [288], and the membrane glutamine receptor (mGlu5) is unable to raise cytosolic Ca^2+^ in conjunction with its function to generate IP_3_ [289] as would be seen during glucose stimulation [290]. Therefore, the role of IP_3_R-mediated Ca^2+^ release from the insulin SG should be reevaluated.

On the inside, both CgA and CgB can interact with the intraluminal loop of all three IP_3_R subtypes at pH 5.5 to stabilize IP_3_ binding and channel rigidity [147,287,291,292,293]. These granins possess an extremely high capacity to bind Ca^2+^ with low affinity at the acidic pH of the MSG [132,146], and it has been suggested that IP_3_R conformational changes can alter that of CgA and CgB to release bound Ca^2+^ through the IP_3_R channel [293,294]. The relative abundance of IP_3_R isoforms and their tetrameric composition as well as that of the interacting granin species are thought to underlie SG Ca^2+^ balance, such that equal amounts of IP_3_ can stimulate varying amounts of Ca^2+^ release [295]. Therefore, the distribution of components within the SG during maturation, idling, and in primed states could modulate SG Ca^2+^ release. Indeed, CgB can undergo redistribution to the MSG periphery upon glucose stimulation [157]. Since intragranular acidification is essential for chromogranin-IP_3_R interactions, it appears that the utility of the SG as an IP_3_-sensitive Ca^2+^ store is reliant on its proper maturation. In conclusion, a host of components are responsible for handling Ca^2+^ for use both inside and outside of the SG.

*Other transporters*. SGs in the β-cell also contain a truncated form of the NHE1 Na^+^/H^+^ exchanger [296], fatty acid translocase (FAT/CD36) [297,298], and a vesicular nucleotide transporter (VNUT) [299]. A role for NHE1 has not yet been determined. CD36, which is predominantly expressed on the PM, has multiple roles throughout the body [300], with a general cellular function to facilitate fatty acid uptake. In β-cells, CD36 is localized to the PM and the SG and mediates the acute and chronic effects of free fatty acids on insulin secretion [297]. CD36 is upregulated in the islets of obese humans with T2D, where altered lipid handling may impair the action of exocytotic proteins [298]. VNUT is a transporter required for ATP uptake into the SG lumen [299] and its depletion results in reduced basal and glucose-stimulated ATP release and insulin secretion [299]. In addition, a Golgi-localized magnesium transporter NIPAL1 was recently shown to positively regulate insulin content and secretion in MIN6 cells [301], albeit through an unknown mechanism.

The activity of channels and transporters provide overarching control on SG maturation and serve as conduits that alter luminal responses based on events transpiring in the cytosol. A prime example is the ability of glucose to drive V-ATPase subunit association to enhance acidification. Together, their presence transforms the SG into a dynamic hub used for ionic signaling and buffering.

### 2.4. Sorting Receptors

Early efforts centered around identifying a ‘*sorting by entry*’ receptor for proinsulin proved futile, possibly because soluble secretory cargo destined for the β-cell MSG are not sorted through receptor-mediated recognition. CPE was once entertained as a candidate for proinsulin sorting from the TGN [302,303], but this was contested [304]. Conversely, the mannose 6 phosphate receptor (M6PR) has been identified as a critical component of ‘*sorting by exit*’ which serves to refine the SG during maturation. It functions to target proteins modified with a mannose 6 phosphate carbohydrate group from the TGN and the ISG to the lysosome, as well as from the plasma membrane for endosomal retrieval [32,305,306,307]. The M6PR specifically contributes to the sorting of the luminal lysosomal hydrolase pro-cathepsin B in β-cells [7,32] (Figure 7).

Two forms of the M6PR exist, a cation independent (CI) ~300 kDa isoform and a ~46 kDa isoform that dimerizes and requires cationic binding (CD—cation dependent) to function. The CD-M6PR locates to the TGN and the ISG where it resides in proximity to clathrin-coated patches and is found on small transport vesicles but not in the MSG [32]. Islets lacking the CD-M6PR contain four-fold more cathepsin B in the β-cell MSG [7]. While the CD-M6PR has a defined role in cathepsin B sorting, a role for the CI-M6PR in the β-cell SG has not yet been determined. To conclude here, little is known about other luminal components that are actively removed from the ISG.

## 3. Concluding Remarks

At its essence, insulin synthesis requires the entry of proinsulin and its processing enzymes into ISGs, followed by enzyme activation and then the complete execution of enzymatic activity. Each of these steps are facilitated by ions that are supplied by channels and transporters, which exert influence on luminal proteins by altering their behavior. First, ion-facilitated granin aggregation and binding to distinct lipid rafts of the TGN membrane may provide the luminal driving force to generate SGs, forming what is currently an ill-defined path of entry for protein to flow through. This is evidenced by observations that granin depletion can limit the biogenesis of nascent SGs. Several resident enzymes of the SG also aggregate specifically in the TGN and bind to lipid rafts to provide a common pathway for sorting, however the trafficking route that key proteins such as proinsulin and proIAPP take are yet to be fully defined. *Sorting by entry* into the ISG is efficient but may not be entirely specific, so the presence of sorting receptors and active ion uptake transporters provide the ISG with two simultaneous quality control mechanisms for refinement—*sorting by exit* and *sorting by retention*. Here, proteins specifically recognized by sorting receptors, and some unwitting bystanders, are escorted from the SG via small transport vesicles, but MSG-destined proteins will bind luminal ions to reduce their solubility and prevent their egression via these carriers. It is currently unclear how important the two post-Golgi sorting mechanisms are for generating insulin itself, but irregular SG maturation could impair overall SG composition and function and encourage the pathological formation of hIAPP oligomers. As the lumen is progressively modified by continual ion uptake, enzymes will begin to exert activity on proinsulin and other peptides to form the complete intragranular cocktail, which will develop into an extremely insoluble ion-bound crystal core surrounded by a halo of soluble components. Here, human T2D-associated Znt8 variants are the exemplar of how ionic alterations can affect parameters of SG maturation, and conceptually illustrate how seemingly small differences can precipitate the chronic disease. Once mature, the SG functions as an intracellular signaling compartment in addition to its role in holding and releasing insulin.

In T2D there is a loss of insulin SGs, GSIS is impaired, and the secretion of proinsulin and its processing intermediates is elevated. Recent subcellular evidence links these phenotypes to a diversion away from SG maturation toward premature ISG secretion, suggesting that MSG formation is a primary limiting factor for insulin secretion in T2D. It is therefore conceivable that defects within the secretory pathway could predispose individuals to the disease by creating an upstream bottleneck to MSG production, delaying the synthesis of MSGs. This might be fine in the healthy state when production is not limiting but could compromise secretion when insulin content is depleted during the chronic condition.

Currently, there is an abundance of knowledge about the distal stages of exocytosis, but a gap in our understanding of events that occur through the late Golgi and the maturing SG. Here, we have provided a comprehensive summary of what happens inside the lumen during the formative hours of the insulin SG. In doing so, it becomes clear that generating a SG that is rich in insulin is quite arduous, and therefore, prone to perturbations that may affect the capacity of the β-cell to adapt to chronic demand.

## Figures and Tables

**Figure 1 metabolites-11-00515-f001:**
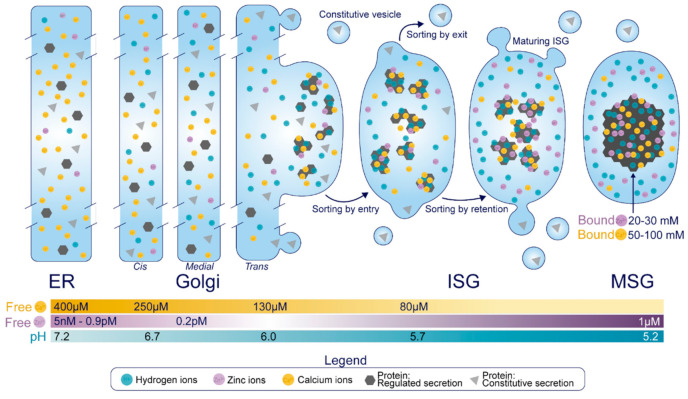
Overview of the β-cell secretory pathway. Following synthesis in the ER, proteins transit the Golgi apparatus to the TGN, and those destined for the regulated secretory pathway are sorted by entry into ISGs. This event relies on soluble protein aggregation which is under the control of Ca^2+^ and H^+^, and several proteins may also interact with membrane components of the TGN that are enriched in β-cell SGs. Some non-SG proteins can also slip into ISGs but are removed as a byproduct of the sorting by exit mechanism, which specifically escorts proteins from the maturing SG via receptor mediated recognition and vesicle budding. Concurrently, Zn^2+^, Ca^2+^, and H^+^ taken up by the maturing SG will bind to certain proteins to enhance their condensation and prevent their exit, in a process termed sorting by retention. While the free concentration of Ca^2+^ is in the micromolar range and decreases proximal to distal along the secretory pathway, the β-cell SG holds 50–100 mM Ca^2+^ bound to luminal proteins. Similarly, the total amount of Zn^2+^ bound to luminal proteins in the SG is in the range of 20–30 mM, although its free concentration is elevated in the distal secretory pathway relative to the proximal secretory pathway. Finally, the pH of the newly formed ISG can be estimated as similar to that of a constitutive vesicle (~5.7), but this will drop to 5.2 in the MSG. Notably, these values represent the free H^+^ concentration, but there exists no indication of the amount of H^+^ that is bound to luminal proteins.

**Figure 2 metabolites-11-00515-f002:**
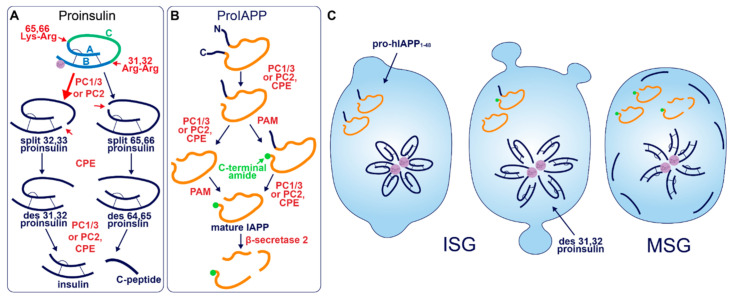
Prohormone Processing in the β-cell. (**A**) Sequence of proinsulin processing. After entry into the ISG, proinsulin is converted to insulin and C-peptide via cleavage at two sites of dibasic amino acid residues. The 31–32 Arg-Arg site is located at the C-peptide/B-chain junction and the 64–65 Lys-Arg site is located between the C-peptide/A-chain junction. Cleavage at one dibasic site by endoprotease PC1/3 or PC2 produces the split proinsulin molecules, which precedes C-terminal trimming of exposed residues by exoprotease CPE to produce the des proinsulin molecules. One round of endo/exoprotease activity is followed by the same action at the other dibasic site. (**B**) Sequence of proIAPP processing. The C-terminal proregion of proIAPP is cleaved in the TGN prior to ISG entry. Next, in no particular order, within the maturing ISG the N-terminal proregion of proIAPP is removed and the exposed C-terminal glycine residue is amidated to produce IAPP. IAPP may then be further processed into smaller fragments by β-secretase 2. (**C**) Processing events and products during secretory granule maturation in the human β-cell SG. des 31,32 proinsulin is the major proinsulin intermediate in human β-cells and is elevated in the circulation of those with T2D along with proIAPP_1–48_.

**Figure 3 metabolites-11-00515-f003:**
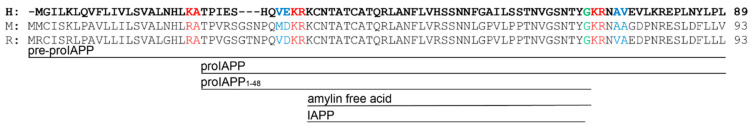
IAPP precursor/product amino acid sequence in human (H), mouse (M), and rat (R). The green glycine residue is amidated after the C-terminal cleavage site is processed. Red residues denote dibasic sites of endoproteolytic processing. Blue residues indicate a modified sequence between species at cleavage sites that could account for their differential specificity to PC enzymes.

**Figure 4 metabolites-11-00515-f004:**
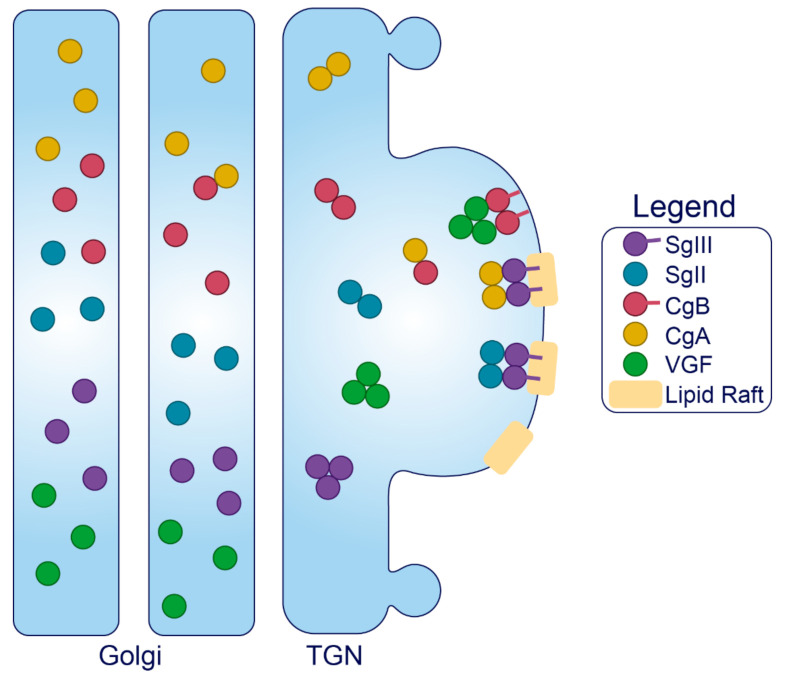
Trafficking of the granins. Upon exposure to mild acidity (pH < 6.4) within the Golgi apparatus, the granins will bind Ca^2+^, which triggers their aggregation and interaction with other granin members. In the TGN, several of these members will interact with target molecules in the membrane to drive the formation of SGs at distinct sites from within the lumen.

**Figure 5 metabolites-11-00515-f005:**
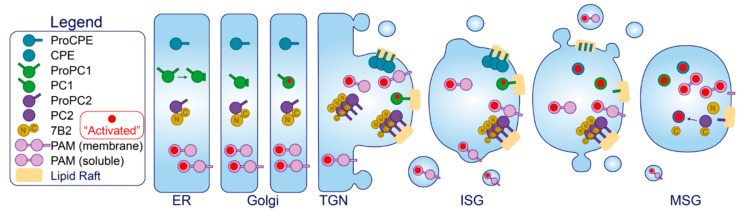
Enzyme suppression and activation. The major processing enzymes of the insulin SG are synthesized as inactive precursors in the ER. Through mechanisms unique to each member, their activity (indicated in red) is suppressed during transit. Activation is principally driven by ionic changes; several enzymes require certain conditions for trafficking into the ISG and conformational activation, and all members require a low pH for optimal enzymatic activity. This will naturally play out through the TGN and the maturing SG as the luminal composition is modified.

**Figure 6 metabolites-11-00515-f006:**
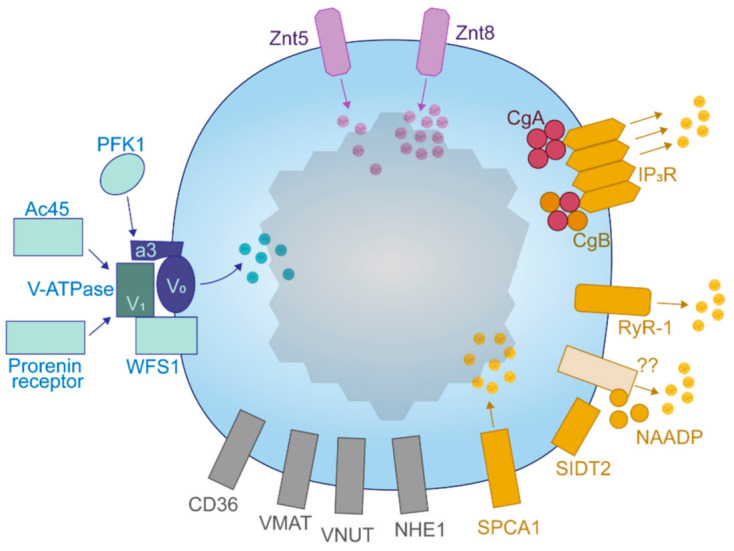
Channels and transporters of the insulin SG. An array of transmembrane components controls the luminal composition of the insulin SG and transform the SG into a responsive store that is utilized by the beta cell for cytosolic signaling.

**Figure 7 metabolites-11-00515-f007:**
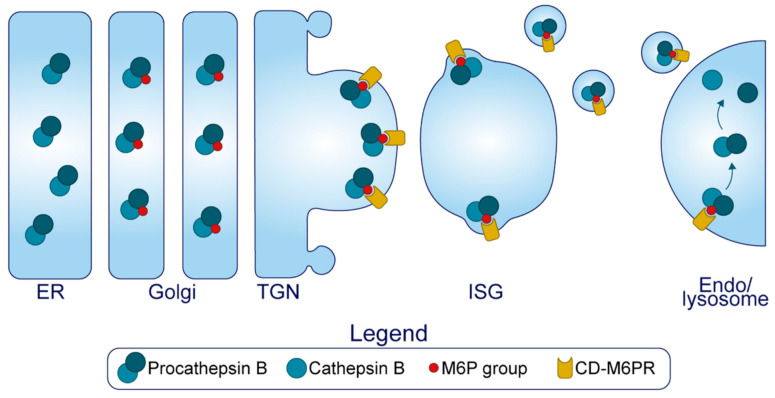
Granule refinement. The cation dependent mannose 6 phosphate receptor (M6PR) binds to certain proteins modified with a mannose 6 phosphate group (namely, procathepsin B) in the TGN, and this complex enters the ISG. During maturation, this complex will exit the SG via small transport vesicles and traffic to the endolysosomal system. A byproduct of this process may be the removal of soluble components in the general vicinity of the budding vesicle.

## Data Availability

Not applicable.

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
