# Peer review of "Inside the Insulin Secretory Granule"

_metabolites, 2021, doi:10.3390/metabo11080515_

Round 1

Reviewer 1 Report

This is an interesting and well written review on the mechanisms regulating insulin synthesis and release in the β-cells. Since it contains a lot of information, it would be advisable that the authors add a brief summary (a few lines) at the end of each major section, as they have done for example for granins (lines 461-463). This will facilitate the readers.

Reviewer 2 Report

Thanks for reviewing and cosidering all downstream aspects invloved in insulin secretion. However, iw was great, but not necesserily as a major comment, to have a look into causative pathogenic conditions such as inflammatory and glucolipotoxicity stresses changing basal and post-prandial insulin biosynthesis and granules. 

Reviewer 3 Report

The manuscript by Germanos et. al. reviews internal processes in insulin secretory granules which lead further to the secretion of mature insulin along with other molecules. Most of the current literature focuses on insulin secretory pathways, i.e. fusion of mature granules with membrane, thus an overview of luminal regulations in secretory granules of beta cells is needed. The topic is carefully reviewed, figures are adequate to the text. There are only a few typos in the text. However, I have few questions which would be great to be introduced in the text.

  1. The content of beta-cell secretory granules (SG) is composed of several molecules besides insulin, such as serotonin, dopamine etc. They are not used for exocytosis events. Can the authors briefly describe their signification for beta cells or further signaling from beta cells?
  2. In T2D and prediabetes, an increased amount of proinsulin is secreted. Does proinsulin have the same signaling role as insulin? What is its role in response to increased blood glucose levels?
  3. The authors describe the essential role of Zn2+ in the crystallization of insulin, proinsulin sorting, and SG maturation. Is it known how obesity, excess of nutrients, HFD, hyperglycemia etc. leading to T2D regulate Zn2+ levels?
  4. Is it known if PC/CPE and other luminal enzymes of SG are regulated by glucose signaling?
  5. Rodents possess 2 insulin genes. Is there any difference in its processing in SG?
  6. Is the luminal composition of ISG the same as MSG in terms of other molecules, enzymes, and IAPP? Is IAPP processing directly linked to proinsulin maturation?
  7. Granin aggregation and binding to lipid rafts have to be affected by membrane composition. What is the role of HFD, nutritional excess in this process?
